# Genetic determinants of zinc homeostasis and its role in cardiometabolic diseases

Marie C. Sadler[1,2,3†], Jean-Pierre Ghobril[1†], Oleg Borisov[4†], Maïwenn Perrais[5,6†], Guglielmo Schiano[7,8†], Dusan Petrovic[9], Eunji Ha[10], Belén Ponte[11], Yong Li[4], Zulema Rodriguez-Hernandez[4], Menno Pruijm[12], Daniel Ackermann[13], Idris Guessous[14], Silvia Stringhini[14], Georg Ehret[15], Tanguy Corre[1], Bruno Vogt[13], Pierre-Yves Martin[11], Halit Ongen[16], Emmanouil Dermitzakis[16], Janet E. Williams[17], Brenda M. Murdoch[17], Michelle K. McGuire[18], Courtney L. Meehan[19], INSPIRE Consortium[¶], Sébastien Lenglet[5], Katalin Susztak[10], Julien Vaucher[20,21‡], Aurélien Thomas[5,6‡], Olivier Devuyst[7‡], Anna Köttgen[4,22‡], Murielle Bochud[1,21‡*], Zoltán Kutalik[1,2,3‡*]

1 Department of Epidemiology and Health Systems (DESS), University Center for General Medicine and Public Health (UNISANTE), Lausanne, Switzerland, 2 Department of Computational Biology, University of Lausanne, Lausanne, Switzerland, 3 Swiss Institute of Bioinformatics, Lausanne, Switzerland, 4 Institute of Genetic Epidemiology, Medical Center – University of Freiburg, Faculty of Medicine, University of Freiburg, Freiburg, Germany, 5 Unit of Forensic Toxicology and Chemistry, CURML, Lausanne and Geneva University Hospitals, Lausanne, Geneva, Switzerland, 6 Faculty Unit of Toxicology, CURML, Lausanne-Geneva, Faculty of Biology and Medicine, University of Lausanne, Lausanne, Switzerland, 7 Institute of Physiology, University of Zurich, Zurich, Switzerland, 8 Functional Genomics Center Zurich, ETH Zurich and University Zurich, Zurich, Switzerland, 9 ClinSearch, Malako, France, 10 Department of Medicine/Nephrology University of Pennsylvania, Perelman School of Medicine, Philadelphia, Pennsylvania, United States of America, 11 Department of Nephrology and Hypertension, Geneva University Hospitals (HUG), Geneva, Switzerland, 12 Department of Nephrology and Hypertension, Lausanne University Hospital (CHUV), Lausanne, Switzerland, 13 University Clinic for Nephrology and Hypertension, Bern University Hospital and University of Bern, Bern, Switzerland, 14 Department and Division of Primary Care Medicine, Geneva University Hospitals, Geneva, Switzerland, 15 Department of Cardiology, Geneva University Hospitals (HUG), Geneva, Switzerland, 16 Genetic Medicine and Development, University of Geneva Medical School-CMU, Geneva, Switzerland, 17 Department of Animal, Veterinary and Food Sciences, University of Idaho, Moscow, Idaho, United States of America, 18 Margaret Ritchie School of Family and Consumer Sciences, University of Idaho, Moscow, Idaho, United States of America, 19 Department of Anthropology, Washington State University, Pullman, Washington, United States of America, 20 Department of Internal Medicine and Specialties, Division of Internal Medicine, Fribourg Hospital and University of Fribourg, Fribourg, Switzerland, 21 Department of Medicine, Division of Internal Medicine, Lausanne University Hospital and University of Lausanne, Lausanne, Switzerland, 22 Centre for Integrative Biological Signalling Studies, Albert-Ludwigs-Universität Freiburg, Freiburg, Germany

¶ A list of authors and affiliations for the INSPIRE Consortium is listed in the S1 File.
† Joint first authors.
‡ Joint last authors.
* murielle.bochud@unisante.ch (MB); zoltan.kutalik@unil.ch (ZK)

## Abstract

Zinc is essential for many physiological processes and its deficiency is highly prevalent worldwide. Its complex homeostasis involves membrane transporters from the SLC39/ZIP and SLC30/ZnT protein families. We conducted a genome-wide association study (GWAS) meta-analysis of urinary zinc levels in three European-ancestry cohorts (N = 10,113), followed by *in silico* and *in vivo* studies to elucidate their underlying public health and physiological relevance. We identified eleven genome-wide

**Data availability statement:** Data availability for SKIPOGH: The SKIPOGH data are available to qualified external researchers upon reasonable request, subject to approval by the SKIPOGH Ethics Committees, to protect participant confidentiality and comply with Swiss data protection regulations. Information on SKIPOGH variables, metadata, and data request procedures is available in the Maelstrom Catalogue (https://www.maelstrom-research.org/study/skipogh). Data availability for CoLaus|PsyCoLaus: The CoLaus|PsyCoLaus data used in this article cannot be fully shared as they contain potentially sensitive personal information on participants. According to the Ethics Committee for Research of the Canton of Vaud, unrestricted sharing would violate Swiss privacy legislation. However, coded individual-level data that do not allow participant identification are available upon request to qualified researchers through the CoLaus | PsyCoLaus Datacenter (CHUV, Lausanne, Switzerland). Applications may be submitted to research.colaus@chuv.ch or research.psycolaus@chuv.ch and will be evaluated by the Scientific Committee. Instructions for data access are available at https://www.colaus-psy-colaus.ch/professionals/how-to-collaborate. Data availability for GCKD: Individual-level data from the GCKD study can be made available to approved collaborators upon request to the GCKD data access committee, in accordance with German data protection laws and participant consent agreements. Details are available at https://www.gckd.org/. Data availability for UK Biobank: Genetic and phenotypic data from the UK Biobank were accessed under application 16389. Data are available to bona fide researchers through the UK Biobank Access Management System (http://www.ukbiobank.ac.uk/using-the-resource/) following institutional approval of a research proposal referencing this project ID. Data availability for INSPIRE: Individual-level, coded data from the INSPIRE cohort used in this article are provided as supplementary material in a CSV-formatted file. Data from the INSPIRE study can be made available to approved collaboration proposals. Please send requests to Dr. Shelley McGuire (smcguire@uidaho.edu) or Dr. Courtney Meehan (cmeehan@wsu.edu), co-principal investigators of the INSPIRE study or famcon@uidaho.edu which is the study

significant signals with six mapping to SLC39/ZIP and SLC30/ZnT gene regions. The lead signal (rs3008217C>G, p = 2.42E-110) in the *SLC30A2* gene region which explained 6.1% of urinary zinc variation strongly colocalized with its expression in kidney tubules. Low phenotypic and genetic correlations between plasma and urinary zinc levels indicated distinct genetic regulation. High urinary zinc correlated with an unfavorable cardiometabolic profile, and Mendelian randomization analyses suggested causal roles for diabetes increasing urinary zinc levels, and elevated urinary zinc increasing stroke risk. Analyzing country-level allele frequencies and zinc deficiency prevalences revealed a 3-fold higher genetic zinc excretion risk in sub-Saharan Africa compared to Europe, significantly correlating with nutritional zinc deficiency prevalence. Although mutations in *SLC30A2* are linked to insufficient zinc in human milk, we found no association with common variants using data generated from 387 mothers. Mice experiments showed that dietary zinc deficiency decreased urinary but not plasma zinc levels, and upregulated kidney *Slc30a2* expression. This first GWAS on urinary zinc highlights the involvement of zinc transporters in its genetic regulation, as well as its role as a non-invasive biomarker for cardiometabolic diseases.

## Author summary

Zinc is the second most abundant trace element in the human body and plays a key role in processes such as immune function and growth. However, zinc deficiency remains common worldwide, particularly in regions with limited access to nutritious food. In this study, we explored the genetic factors that influence urinary zinc excretion and their health implications. By analyzing genetic data from over 10,000 individuals, we identified several genetic variants associated with urinary zinc levels, many of which reside in genes encoding major zinc transporter proteins. We also found that higher urinary zinc levels were linked to a less favorable cardiometabolic profile. Genetic variants associated with higher zinc excretion were more frequent in individuals from sub-Saharan Africa, where zinc deficiency is also more prevalent. In mice, we confirmed that dietary zinc deficiency reduced urinary but not plasma zinc levels. Together with our findings that different genes regulate zinc in blood and urine, our results suggest that the body adjusts kidney function to stabilize circulatory zinc levels, highlighting urinary zinc as a potentially more sensitive marker of zinc status and metabolic health.

## Introduction

Zinc is the second most abundant trace element in the human body and is essential for major physiological processes such as organ development, growth, immuno-competence, or neurobehavioral function [1,2]. The biological relevance of zinc is

co-ordination center.. (Citation: McGuire, S.M., Meehan, C.L. et al., The INSPIRE Study: What's normal? Oligosaccharide concentrations and profiles in milk produced by healthy women vary geographically. Am J Clin Nutr. 2017;105: 1086–1100.) Publicly available summary data: The zinc meta-GWAS summary statistics is available in the GWAS Catalog under the accession ID GCST90672046. Availability of summary statistics used for the colocalization analyses is described in the Method section "Omics integration and colocalization analyses". Neale's lab GWAS summary statistics (UK Biobank) are available at http://www.nealelab.is/uk-biobank. Type 2 diabetes GWAS summary statistics (meta-analysis of DIAGRAM consortium and UK Biobank) is available at: http://diagram-consortium.org/. Coronary artery disease GWAS summary statistics (meta-analysis of CARDIoGRAMplusC4D and UK Biobank) is available at: https://www.cardiogramplusc4d.org/data-downloads/. Stroke GWAS summary statistics (MEGASTROKE Consortium) is available at: https://www.megastroke.org/ Estimated zinc deficiency prevalences from zinc availability in national food supplies are available at: Table S2 from Wessels et al. 2012 at https://doi.org/10.1371/journal.pone.0050568 . Measured zinc deficiency prevalences based on plasma/serum zinc concentration (PZC) in Low- and Middle-income Countries: Table 1 from Gupta et al., 2020 at DOI: 10.1111/jhn.12791.

**Funding:** The SKIPOGH study is supported by the Swiss National Science Foundation (Grants: FN33CM30-124087 and FN33CM30-140331). ZK was funded by the Swiss National Science Foundation (310030-189147, 315230-219587). AT was funded by the Swiss National Science Foundation (31003A-182420). The CoLaus|PsyCoLaus study was supported by research grants from GlaxoSmithKline, the Faculty of Biology and Medicine of Lausanne, the Swiss National Science Foundation (grants 3200B0–105993, 3200B0-118308, 33CSCO-122661, 33CS30-139468, 33CS30-148401, 33CS30-177535, 31003 A-182420, 3247730-204523 and 320030-220190) and the Swiss Personalized Health Network (grant 2018DRI01). The ToxiLaus study was supported by a grant from the Fondation pour la recherche sur le diabète (https://fondation-diabete.ch). OD and GS were supported by the European Reference Network for Rare Kidney Diseases (project N° 739532), the

explained by its structural incorporation within >10% of all human proteins, including transcription factors, structural proteins, and enzymes, which are involved in various cellular functions such as gene expression, DNA repair, protein metabolism, or cell signaling [3–6]. While zinc is ubiquitously present in the human habitat, it is only available for human metabolism via dietary intake. Its main dietary sources are meat, fish, dairy, legumes, and whole grain cereals, with a greater bioavailability in animal products than in plant-based foods, as phytate compounds in plants tend to chelate zinc and reduce its absorption in the human gut [1,5,7,8]. Histidine and methionine, as well as EDTA and organic acids increase its absorption [9]. Zinc content in food is directly linked to zinc content in agricultural soils. Several studies have highlighted the relationship between dietary zinc, nutritional zinc status and soil zinc content [10–12].

Through malabsorption, malnutrition or zinc losses, zinc deficiency leads to serious health conditions including growth stunting, hypogonadism, impaired immune response, skin lesions, and neurocognitive disorders [13]. While severe zinc deficiencies are rare and usually caused by genetic defects, mild-to-moderate zinc deficiencies remain prevalent in up to 20% of the worldwide population and result from an insufficient zinc intake [1,7,13,14]. In particular, it has been consistently reported that mild-to-moderate nutritional zinc deficiencies are responsible, among others, for anorexia, diarrhea, impaired immune system, night blindness or mild psoriasiform dermatitis, principally in Africa, the Eastern Mediterranean, and South-East Asia [13,15].

Due to its central role in human physiology, zinc homeostasis is tightly controlled through the action of >20 membrane transporters belonging to two large families, the Zrt/Irt-like (SLC39/ZIP) protein family, which import $Zn^{2}+$ ions into the cytosol from the extra/intracellular compartments, and the Zinc Transporter (SLC30/ZnT) family, which export $Zn^{2}+$ out of the cytosol [6,16]. Rare genetic defects in these transporters lead to serious health conditions in pediatric populations, including severe dermatitis, chronic diarrhea, alopecia, growth stunting, and even death if adequate zinc supplementation is not provided [14,17,18]. Loss-of-function mutations in the *SLC39A4*/*ZIP4* gene lead to acrodermatitis enteropathica, an autosomal-recessive disorder caused by a decreased intestinal absorption of zinc, resulting in severe, lifelong zinc deficiency [18]. Heterozygous mutations in the *SLC30A2*/*ZnT2* gene are associated with transient neonatal zinc deficiency (TNZD), characterized by an insufficient secretion of zinc into human milk causing zinc deficiency in breastfed infants [14,16,18].

While the physiological role of zinc and its transporters has been increasingly documented in the scientific literature, a population-based, genome-wide investigation of genetic variants and their functional role associated with physiological zinc levels is under-researched. A recent study investigating the genetic determinants of zinc levels in whole blood identified three associated loci (near genes *PPCDC, CA1* and *LINC01221*) [19]. While circulating zinc levels are physiologically important, they are not a reliable indicator of individual zinc status [20]. Plasma zinc, recognized as the most reliable and widely utilized biomarker of population zinc status, is sensitive

Swiss National Science Foundation (grant 10.003.608), and the University Research Priority Program (URPP) ITINERARE at the University of Zurich. The INSPIRE study was supported by grants from the National Science Foundation (DBI-0939454, 1344288, 1917476), Washington State University Office of Research GrandChallengeNutritional Genomics Initiative, and the USDA National Institute of Food and Agriculture (Hatch projects IDA01643 and IDA01566). The work of OB and AK was funded by German Research Foundation (DFG) project ID 431984000 (SFB 1453). The work of YL and AK was supported by Germany's Excellence Strategy (Centre for Integrative Biological Signalling Studies, EXC-2189, project ID 390939984). The work of OB was funded by the Hans A. Krebs Medical Scientist Program, Faculty of Medicine, University of Freiburg. The work of ZRH was supported by a fellowship associated to a National Research Agency project (PRE2020-093926; PID2019-108973RB-C21) funded by Ministerio de Ciencia e Innovacion (Spain), an EMBO Scientific Exchange Grant (number:10351, 2023), and by the German Research Foundation (DFG) Project-ID 530592017 (SCHL 2292/3-1). The GCKD study was and is supported by the BMBF (FKZ 01ER 0804, 01ER 0818, 01ER 0819, 01ER 0820 and 01ER 0821) and the KfH Foundation for Preventive Medicine. Unregistered grants to support the study were provided by corporate sponsors (listed at https://gckd.org). The funders had no role in study design, data collection and analysis, decision to publish, or preparation of the manuscript.

**Competing interests:** I have read the journal's policy and the authors of this manuscript have the following competing interests: MCS has been consulting for 5 Prime Sciences at the time of the submission; however, this study was performed separately with no relationship to 5 Prime Sciences. The results and opinions expressed in this paper do not represent those of 5 Prime Sciences. The other authors declare that they have no competing interests.

to severe zinc deficiency; however, it has limited capacity to detect mild to moderate deficiency states [21]. Although previous studies suggested that plasma zinc increases with dietary supplementation [22,23], the overall picture is more complex. An analysis of NHANES data shows that zinc concentrations are more related to sex, age, serum albumin and hemoglobin levels as well as the time of blood draw than to dietary zinc intake or zinc supplements [24]. Studies assessing zinc enrichment in largely consumed food also show no change in plasma zinc concentration despite improvement in selected growth indicators in children [25,26]. Zinc is involved in numerous essential cellular processes affecting gene expression, enzymatic functions and structural aspects in all compartments of the human body [27]. These findings are supported by data showing that only about 1.5% of body zinc is found in the blood compared with bone (36.7%) and muscle tissue (49.5%), where zinc is bioavailable but not readily mobilizable [27].

Given the important role of the kidneys in regulating systemic levels of many trace elements, investigating zinc levels in urine offers a deeper insight into zinc homeostasis [28]. Thus, to understand the genetic determinants of urinary zinc excretion and zinc deficiency, we conducted a genome-wide association study (GWAS) of urinary zinc levels in three prospective cohorts and investigated shared molecular pathways potentially modulating urinary zinc levels in an omics-wide colocalization framework. We contrasted epidemiological and genetic associations using bidirectional Mendelian randomization (MR) and further demonstrated stark differences between the genetic underpinnings of blood *vs* urinary zinc levels. Moreover, we explored the relation between the allele frequency of the significantly associated genetic variants and the prevalence of zinc deficiency in multiple human populations across the globe by applying an ecological correlation analysis. To better understand the underlying mechanisms between the strongest associated variant, impacting *SLC30A2* gene expression, we used an animal zinc feeding model to characterize the biological function of SLC30A2/Znt2 in zinc homeostasis. Lastly, we examined whether the identified genetic associations with zinc levels were mirrored in human milk (Fig 1).

## Results

### Genetic loci associated with urinary zinc excretion

We conducted a GWAS of urinary zinc levels in three prospective cohorts - SKIPOGH (Swiss Kidney Project on Genes in Hypertension, N = 896) [29,30], CoLaus|PsyCoLaus (Cohort Lausanne, N = 4,542) [31] and GCKD (German Chronic Kidney Disease study, N = 4,822) [32] - for a total of 10,113 participants (Figs 2, A and B in S1 File and Tables 1 and 2). We identified 11 independent genome-wide significant (p-value < 5E-08) signals at nine genomic loci. Six of these independent signals at four loci map to gene regions containing members of the two large zinc transport families SLC39/ZIP and SLC30/ZnT (*SLC30A2, SLC30A1, SLC39A8* and *SLC39A4*). The lead association signal in the *SLC30A2* gene region (rs3008217G>C, beta = 0.222 SD, p-value = 2.42E-110) also displayed strong effects when separating day, night and 24h urinary zinc excretion, as well as the fractional excretion of zinc, which were available in the SKIPOGH cohort. Each additional C allele consistently displayed

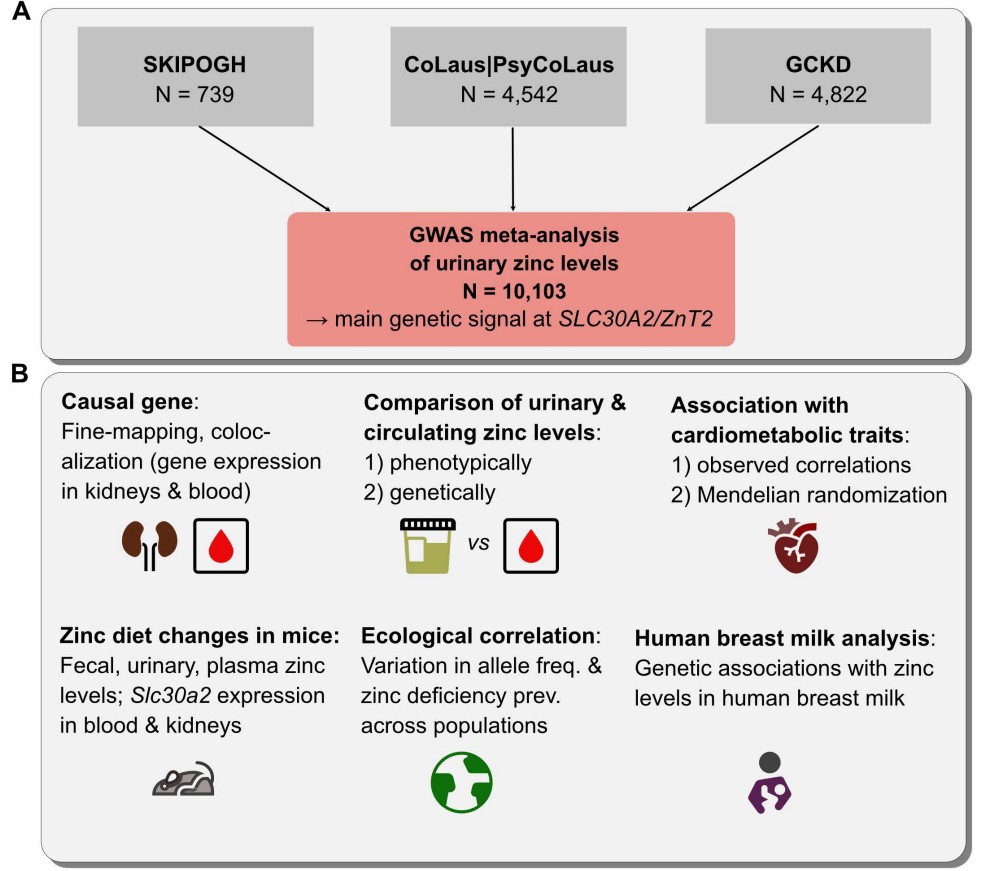

**Fig 1. Overview of the analysis. Panel A:** A genome-wide association study (GWAS) meta-analysis of urinary zinc levels was performed on 10,113 individuals of European descent across three prospective cohorts. **Panel B:** Downstream analyses searched for the most likely causal genes underlying the identified genetic signals, their biological functions, differences between urinary and circulating zinc levels, and the epidemiological relevance of urinary zinc levels and the genetics thereof in relation to cardiometabolic diseases and global zinc deficiency prevalence.

higher day, night, and 24h urinary zinc excretion levels, and higher zinc excretion fraction when compared to GG and GC genotypes (Fig C in S1 File). Although all cohorts identified a genome-wide significant directionally concordant association with rs3008217, strong heterogeneity in effect sizes was identified ($P_{HET}$=8.12E-12). This heterogeneity was driven by a 45% lower effect of this allele in the GCKD cohort compared to the effects observed in the two other (Swiss) cohorts. GCKD study participants were, on average, older and had lower kidney function compared to the participants from other studies, but the reduced effect in GCKD was not explained by SNP-age or SNP-eGFR interactions ($P_{Gxage}$=0.16, $P_{Gxe-GFR}$=0.69). Note that this locus exhibited three independent association signals as determined by fine-mapping analysis. Three credible sets (CS) were identified and within these sets we identified the SNPs with the highest probability of being causal. This was assessed by estimating the highest posterior inclusion probability (PIP), a value between zero and one, with zero representing no support and one corresponding to 100% certainty. At this locus the following three SNPs were the most likely causal variants: rs11587475 (PIP=0.38, CS=1, 3'UTR variant of *SLC30A2*), rs3008217 (PIP=0.27, CS=2, intronic variant of a lncRNA ENSG00000284309) and rs17257086 (PIP=0.09, CS=3, intronic variant of *TRIM63*). Given the heterogeneity of effect sizes at this locus, we performed fine-mapping in each cohort separately. While rs3008217 was the top SNP in the meta-analysis and CoLaus|PsyCoLaus, GCKD and SKIPOGH identified different lead SNPs with the highest PIP (rs6686190 with PIP=0.14 in GCKD and rs3008208 with PIP=0.25 in SKIPOGH; S1 Dataset). Together

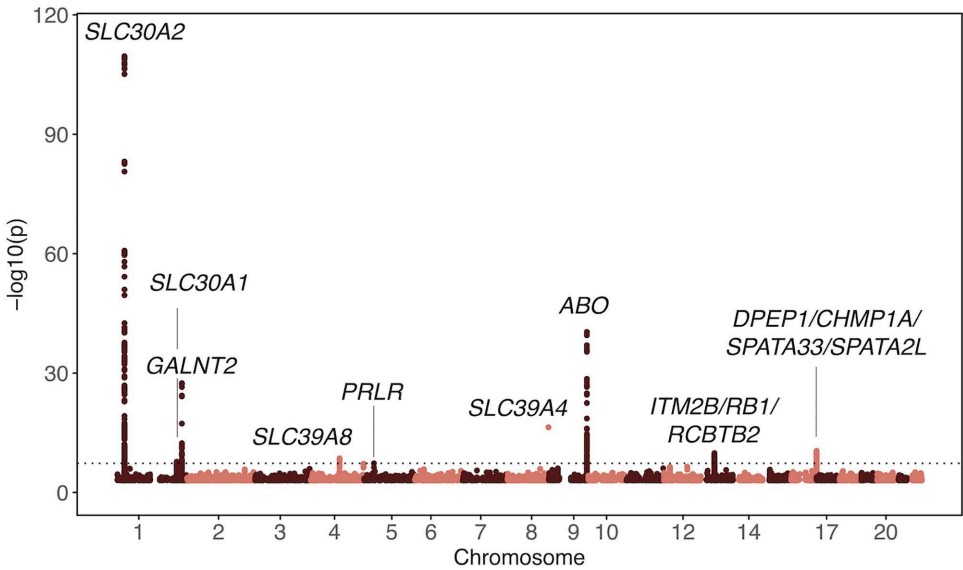

**Fig 2. Manhattan plot of the urinary zinc levels association meta-analysis conducted in three cohorts (N = 10,113).** Genome-wide significant signals (p-value < 5E-08) are annotated with most plausible causal genes identified through fine-mapping and colocalization analyses. The horizontal line denotes genome-wide significance.

**Table 1. Characteristics of the study populations. Urinary zinc levels are adjusted for creatinine levels to account for urine concentration. BMI - body mass index; no - number; eGFR - estimated glomerular filtration rate; UACR - urine albumin-to-creatinine ratio, IQR - interquartile range.**

| Characteristic | SKIPOGH (N = 896) | CoLaus \| PsyCoLaus (N = 4,542) | GCKD (4,822) |
|---|---|---|---|
| Age (Years) Mean (SD) | 47.4 (17.5) | 53.5 (10.7) | 60.1 (12.0) |
| Female sex – no (%) | 53% | 2418 (53.2%) | 1909 (39.6%) |
| BMI (kg/m2) Mean (SD) | 25.1 | 25.8 (4.51) | 29.8 (5.95) |
| Smokers - no (%) | 208 (23.2%) | 1217 (26.8%) | 777 (16.1%) |
| Diabetes - no (%) | 38 (4.2%) | 315 (6.9%) | 1696 (35.2%) |
| eGFR (mL/min/1.73m2): Mean (SD) | 95.9 (17.6) | 83.0 (16.4) | 49.5 (18.2) |
| Creatinine (g/L) Mean/Median (SD, IQR) | 1.0/ 0.9 (0.54, [0.6-1.3]) | 1.50/ 1.41 (0.735, [0.996-1.89]) | 1.51/ 1.45 (0.47, [1.19-1.75]) |
| UACR (mg/g) Median (IQR) | 4.34 [2.8-7.5] | 5.1 [3.41-9.05] | 50.5 [9.6-388.3] |
| Urinary zinc levels (µg/g) Mean/ Median (SD, IQR) | 271/ 241 (150, [167-347]) | 365/ 296 (843, [198-434]) | 434.7/316.2 (430.2, [171-552]) |

with the modest PIP of 0.38 in the meta-analysis, this indicates low confidence in rs3008217 being the causal variant and suggests the presence of an alternative, possibly unmeasured, causal SNP driving the SLC30A2 signal.

The second strongest signal in the urinary zinc GWAS was located in the *ABO* gene (rs2519093C>T, beta = 0.133, p-value = 4.63E-41). Remaining hits included genetic variants mapping to the *GALNT2*, *PRLR*, *ITM2B/RB1/RCBTB2* and *DPEP1/CHMP1A/SPATA33/SPATA2L* genetic regions. For all loci other than *SLC30A2*, a single CS was identified with the variant with the highest PIP corresponding to the top SNPs of Table 2 (Figs D-L in S1 File and S2 Dataset).

**Table 2. Significant independent hits with most plausible causal genes. Chr - chromosome; AF – allele frequency. Average imputation qualities (R²) were above 0.9.**

| Chr | Position (GRCh37) | SNP | Mapped Gene | Effect allele | Other allele | Effect AF | N | beta | p-value |
|---|---|---|---|---|---|---|---|---|---|
| 1 | 26122979 | rs807244 | *SLC30A2* | A | G | 0.803 | 10113 | -0.059 | 2.36E-09 |
| 1 | 26373694 | rs3008217 | *SLC30A2* | C | G | 0.120 | 10113 | 0.222 | 2.42E-110 |
| 1 | 26650661 | rs41305769 | *SLC30A2* | A | G | 0.952 | 10113 | -0.064 | 1.25E-10 |
| 1 | 211803881 | rs12734494 | *SLC30A1* | A | G | 0.478 | 10113 | 0.056 | 1.92E-08 |
| 1 | 230266459 | rs61825397 | *GALNT2* | A | T | 0.852 | 10113 | 0.109 | 3.38E-28 |
| 4 | 103198082 | rs13135092 | *SLC39A8* | A | G | 0.914 | 10113 | 0.059 | 2.84E-09 |
| 5 | 35085162 | rs138587215 | *PRLR* | T | C | 0.012 | 9364 | 0.056 | 4.87E-08 |
| 8 | 145639726 | rs2272662 | *SLC39A4* | T | C | 0.426 | 10113 | 0.084 | 4.20E-17 |
| 9 | 136141870 | rs2519093 | *ABO* | T | C | 0.226 | 10113 | 0.133 | 4.63E-41 |
| 13 | 48891836 | rs3825417 | *ITM2B/RB1/RCBTB2* | A | G | 0.783 | 10113 | -0.064 | 1.31E-10 |
| 16 | 89696217 | rs2434860 | *DPEP1/CHMP1A/SPATA2L* | T | C | 0.496 | 9364 | -0.068 | 3.50E-11 |

## Omics integration and mechanistic pathways

We integrated evidence from tissue-specific gene expression information (eQTL) as well as protein expression in whole blood (pQTL) via fine-mapping and colocalization approaches to map identified zinc-associated loci to their most plausible causal genes (S3 Dataset). We next used the "coloc.susie" method to test for colocalization between variants in each of three CS and gene expression, and detected evidence for the same genetic variants underlying the association with zinc levels and the expression of *SLC30A2* in kidney tubules (PP.H4 = 0.99; Fig 3A, Methods). Additionally, we identified colocalization of the zinc levels with *SLC30A2* expression in other tissues from the GTEx data: sun- and non-sun-exposed skin (PP.H4 = 0.86 and PP.H4 = 0.799, respectively) supporting a shared genetic architecture of *SLC30A2* expression and zinc levels in urine. In the following, we only mention the most notable links found, but the full colocalization analysis results can be found in S3 Dataset.

Notably, the third locus on chromosome 1 (lead variant rs61825397) showed colocalization (PP.H4 = 0.992) with *GALNT2* expression in kidney tubules based on the meta-analysis of eQTL datasets (Fig 3B). The locus on chromosome 8 had only one significantly associated SNP (rs2272662) with no other SNPs in linkage disequilibrium (LD) >0.3 (Fig B in S1 File). However, the associated top hit is itself a missense variant of *SLC39A4* strengthening its role in zinc metabolism. The *ABO* locus on chromosome 9 did not colocalize with *ABO* expression in kidney tubules, but we observed colocalization with *ABO* expression in spleen based on the GTEx data. Given the known pleiotropy of *ABO*, we detected colocalization with several hundred proteins (in *trans*) as well as clinical traits, most prominently with cardiovascular ones (coronary heart disease, hyperlipidemia, pulmonary heart disease). We observed colocalization signals with 697 traits in total (PP.H4 > 0.5) and among them 632 traits showed strong colocalization evidence (PP.H4 > 0.8). We also observed colocalization at the *ABO* locus with blood urea nitrogen as a marker of kidney function. Given the extreme pleiotropy of this locus, we cannot conclude any driver gene in particular. To further assess whether the association in the *ABO* region stems from a causal association between ABO blood type and zinc excretion, we consulted the ABO blood type group of 2,478 participants in CoLaus|PsyCoLaus with available measurements. Although we found that individuals with the O-group had lower zinc levels compared to the A, B and AB blood groups ($b_{obs}$ = -0.24; p-value = 1.05E-09), we could not find evidence that the ABO blood type group mediates the genetic association (Note C in S1 File). The region on chromosome 16 colocalized with expression of several genes based on the meta-analysis of kidney tubule eQTL data: *DPEP1*

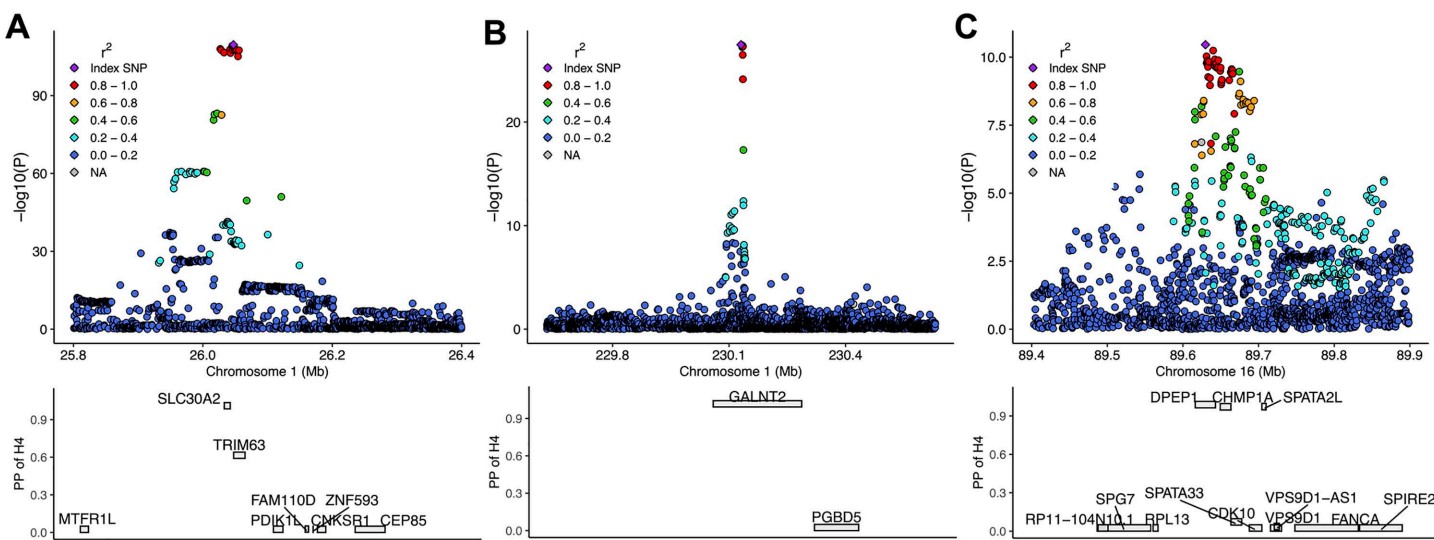

**Fig 3. Regional association plots of genetic associations with urinary zinc levels and colocalization results with gene expression in kidneys. Panel A-B-C**: The top plot shows the genetic variants and their associations with urinary zinc levels (negative logarithm of p-values) color-coded based on linkage disequilibrium (LD) with the lead SNP, highlighted in purple. The bottom plot shows colocalization results for each gene in the region summarized by the posterior probability of hypothesis 4 (PP of H4) for a shared causal variant. A high PP of H4 value suggests evidence of shared causality between expression levels of the gene in kidneys and urinary zinc levels. The x-axis indicates the genomic position and gene location.

(PP.H4 = 0.94), *CHMP1A* (PP.H4 = 0.85), and *SPATA2L* (PP.H4 = 0.9; Fig 3C). In addition, we observed colocalization with gene expression in two additional transcriptomics datasets, *SPATA33* and *CHMP1A*, in eQTLGen and *SPATA33* and *SPATA2L* in GTEx. Proteomics data integration identified colocalization with circulating levels of DPEP1 and CHMP1A in two studies (Methods). Interestingly, for three zinc-associated loci (on chromosomes 4, 8, and 9) we observed colocalization with circulating levels of carboxypeptidases encoded in *trans*, which are zinc-containing metallo-enzymes, e.g., *CPQ* and *CPB1* (S3 Dataset).

## Comparison of urinary and circulating zinc levels

Next, we studied the relationship between urinary and circulating zinc levels. In the SKIPOGH cohort, plasma zinc concentrations were available in addition to urinary zinc levels and revealed a low phenotypic correlation of $\rho_{Pearson} = 0.17$ (p-value = 2.41E-07, this correlation being adjusted for study-specific covariates; Methods). We then compared the genetic architecture of urinary and blood zinc concentration by consulting the largest blood zinc meta-GWAS to date conducted in 6,564 Scandinavian participants of the HUNT, MoBa and PIVUS cohorts [19]. By comparing genetic associations in regions that contained genome-wide significant variants in either the urinary or blood zinc GWAS, we found that only rs13135092 (*SLC39A8*; p-value of 2.84e-09 in the urinary zinc excretion GWAS) and rs2434860 (*DPEP1/CHMP1A/SPATA2L*; p-value of 3.50E-11 in the urinary zinc excretion GWAS) were nominally significant in the blood zinc GWAS with effects in opposite directions (p-values of 8.05E-03 and 0.042, respectively), whereas the remaining SNPs did not show any overlap even at the nominal significance level (Table A in S1 File). This poor overlap in genetic architecture was also supported by a non-significant genetic correlation ($r_g$ = -0.065; 95% CI = [-0.504, 0.373]). Finally, we conducted bidirectional MR analyses to investigate whether increased blood zinc levels were causal for increased urinary blood zinc levels and vice versa. In line with non-significant genetic correlations, we could not identify such a relationship (blood zinc→urine zinc: $b_{MR}$ = 0.024, p-value = 0.51; urine zinc→blood zinc: $b_{MR}$ = -0.018; p-value = 0.71). Overall, these results suggest that zinc homeostasis is controlled by very different genetic factors in blood than in urine.

## Phenotypic correlations between urinary zinc levels and cardiometabolic traits

We studied the epidemiological associations between urinary zinc levels and clinical traits in the CoLaus|PsyCoLaus cohort (Table 3). Accounting for multiple testing (Bonferroni corrected p-value threshold of 0.05/15 = 3.33E-03), higher urinary zinc levels were significantly associated with an unfavorable cardiometabolic profile, namely higher systolic and diastolic blood pressure, total cholesterol and triglyceride levels, as well as increased C-reactive protein (CRP) and glucose levels. These quantitative trait associations were also reflected in the associations with the related disease status, as higher urinary zinc excretion was associated with diabetes (OR = 2.35, p-value = 1.45E-30) and stroke events (OR = 1.21, p-value = 3.91E-03). The association between urinary zinc levels and stroke could also be corroborated in a time-to-event analysis (hazard ratio = 1.25; 95% CI = [1.10, 1.41]; p-value = 4.4E-04). Additionally, significant associations were found between higher urinary zinc levels and anthropometric traits including increased weight (p-value = 2.96E-15), BMI (p-value = 1.01E-22) and waist-to-hip ratio (p-value = 4.65E-24), but shorter stature (p-value = 9.14E-04). Taken together, increased urinary zinc excretion was found to be a marker of impaired health for several cardiometabolic conditions, in addition to being associated with higher weight/BMI but lower height.

## Bidirectional Mendelian randomization results

To examine whether the observed phenotypic associations could mean causal relationships, we performed bidirectional Mendelian randomization analyses. These revealed a (Bonferroni) significant (<0.05/15 = 3.3E-03), but modest, forward causal effect of increased urinary zinc levels on increased stroke risk ($b_{MR}$ = 0.031; p-value = 2.15E-03; Table 4) suggesting that the observed epidemiological association may be driven by a true causal link. Given the high pleiotropy among instruments, mostly caused by the instrumental variables (IVs) associated with *ABO* and *SCL39A8*, we conducted several sensitivity analyses to corroborate the MR associations (Figs M-N and Tables B-C in S1 File). Although the causal effect

**Table 3. Epidemiological associations between urine zinc concentrations and quantitative/disease phenotypes.** Observed effect size - association coefficient (SD/SD) or odds ratio (OR); se - standard error; SBP - systolic blood pressure; DBP - diastolic blood pressure; HDL-C - high-density lipoprotein cholesterol; LDL-C - low-density lipoprotein cholesterol; TC - total cholesterol; TG - triglycerides; CRP - C-reactive protein; Diabetes - diabetes defined as fasting plasma glucose levels above 7.0 mmol/L; CAD - coronary artery disease; WHR - Waist-to-hip ratio; BMI - body mass index.

| Trait/Disease | Observed effect size | se | p-value | Sample Size N (Ncases/controls) |
|---|---|---|---|---|
| SBP | 0.082 | 0.011 | 2.04E-13 | 6397 |
| DBP | 0.070 | 0.012 | 1.43E-08 | 6397 |
| TC | 0.045 | 0.013 | 4.80E-04 | 6394 |
| HDL-C | 0.010 | 0.011 | 3.58E-01 | 6394 |
| LDL-C | -0.007 | 0.013 | 6.18E-01 | 6302 |
| TG | 0.122 | 0.012 | 8.81E-25 | 6394 |
| Glucose | 0.172 | 0.012 | 1.49E-46 | 6394 |
| CRP | 0.050 | 0.012 | 2.54E-05 | 6381 |
| Diabetes | 2.347 (OR) | 0.074 | 1.45E-30 | 419/5975 |
| Stroke | 1.210 (OR) | 0.066 | 3.91E-03 | 295/5368 |
| CAD | 1.115 (OR | 0.058 | 5.75E-02 | 429/5975 |
| Weight† | 0.086 | 0.011 | 2.96E-15 | 6403 |
| WHR† | 0.124 | 0.012 | 4.65E-24 | 6403 |
| BMI† | 0.097 | 0.010 | 1.01E-22 | 6398 |
| Height† | -0.032 | 0.010 | 9.14E-04 | 6403 |

Associations are adjusted for sex, age, smoking status, zinc measurement batch and BMI (phenotypes marked with † are unadjusted for BMI).

**Table 4. Bidirectional Mendelian randomization (MR) results between urine zinc concentrations and quantitative/disease phenotypes.** Forward MR results quantify the causal effect of urinary zinc concentrations on the trait ($b_{MR}$-forward), and reverse MR results the causal effect of the trait on urinary zinc concentrations ($b_{MR}$-reverse). $b_{MR}$ - MR causal effect estimated through the inverse-variance weighting method (SD/SD); $n_{IV}$ - number of instrumental variables; $Q_{HET}$ p-value - p-value from a Cochran's Q-statistic heterogeneity test; $F_{stat}$ - F-statistics quantifying the strength of association between the instrumental variables and the exposure; T2D - type 2 diabetes; CAD - coronary artery disease; SBP - systolic blood pressure; DBP - diastolic blood pressure; LDL-C - low-density lipoprotein cholesterol; TC - total cholesterol; TG - triglycerides; CRP - C-reactive protein.

| Trait | $b_{MR}$ (forward) | p-value$_{MR}$ (forward) | $n_{IV}$ (forward) | $Q_{HET}$ p-value (forward) | $F_{stat}$ (forward) | $b_{MR}$ (reverse) | p-value$_{MR}$ (reverse) | $n_{IV}$ (reverse) | $Q_{HET}$ p-value (reverse) | $F_{stat}$ (reverse) |
|---|---|---|---|---|---|---|---|---|---|---|
| T2D | 0.001 | 9.40E-01 | 11 | 2.09E-15 | 102.3 | 0.135 | 3.38E-04 | 315 | 2.00E-04 | 67.8 |
| HbA1c | 0.029 | 2.94E-01 | 11 | 2.22E-59 | 102.3 | -0.023 | 3.79E-01 | 733 | 2.28E-09 | 91.9 |
| Glucose | 0.015 | 3.72E-01 | 11 | 9.98E-17 | 102.3 | 0.056 | 2.75E-01 | 168 | 1.73E-02 | 89.2 |
| CAD | 0.020 | 1.25E-01 | 9 | 3.53E-09 | 116.9 | 0.027 | 6.13E-01 | 226 | 1.25E-01 | 65.8 |
| SBP | 0.016 | 1.71E-01 | 11 | 3.32E-07 | 102.3 | 0.078 | 9.19E-02 | 374 | 3.51E-02 | 48.3 |
| DBP | -0.001 | 9.76E-01 | 11 | 1.89E-23 | 102.3 | 0.135 | 1.36E-02 | 319 | 1.38E-06 | 50.1 |
| TC | 0.047 | 1.22E-01 | 11 | 1.31E-69 | 102.3 | 0.006 | 8.46E-01 | 391 | 1.91E-05 | 114.3 |
| LDL-C | 0.045 | 1.22E-01 | 11 | 1.65E-62 | 102.3 | -0.017 | 5.82E-01 | 368 | 1.82E-04 | 121.5 |
| TG | -0.034 | 1.59E-02 | 11 | 3.25E-12 | 102.3 | 0.139 | 7.91E-08 | 591 | 4.02E-03 | 102.0 |
| Stroke | 0.031 | 2.15E-03 | 9 | 1.17E-01 | 116.7 | -0.378 | 3.56E-02 | 11 | 6.96E-01 | 41.1 |
| CRP | 0.027 | 9.90E-02 | 11 | 2.77E-18 | 102.3 | -0.053 | 6.06E-02 | 488 | 1.45E-02 | 100.0 |
| Height | -0.021 | 2.76E-01 | 11 | 1.96E-27 | 102.3 | 0.013 | 3.89E-01 | 2294 | 1.91E-02 | 75.7 |
| Bone mineral density | -0.018 | 2.38E-01 | 11 | 3.49E-08 | 102.3 | 0.037 | 9.70E-02 | 581 | 2.94E-01 | 75.5 |
| Calcium | 0.012 | 6.46E-01 | 11 | 3.58E-46 | 102.3 | 0.085 | 9.93E-03 | 415 | 1.44E-01 | 76.0 |
| Birthweight | 0.000 | 9.70E-01 | 11 | 4.67E-04 | 102.3 | -0.027 | 6.80E-01 | 126 | 6.22E-03 | 50.1 |

with stroke did not reach significance when applying the simple mode and median MR methods, likely due to the low number of IVs, the causal effect on stroke remained robust in the MR Egger analysis (p-value = 0.01) and after removing *ABO* and *SCL39A8* associated IVs (Table D in S1 File).

Inspecting reverse causal relationships, we found that increased genetic liability to type 2 diabetes and elevated triglyceride levels, increased zinc excretion ($b_{MR}$ = 0.135, p-value = 3.38E-04 and $b_{MR}$ = 0.139, p-value = 7.91E-08, respectively), which remained significant in the Egger, simple median and mode MR methods (Table E in S1 File). These identified relationships suggest that the epidemiological correlations between urinary zinc levels and type 2 diabetes is a likely indirect consequence of a reverse causal effect, potentially via kidney hyperfiltration resulting from hyperglycemia-associated elevated urine volume.

## Expression of SLC30A2 and dietary zinc modifications in mouse

In order to better understand the molecular mechanisms behind the by-far strongest GWAS association signal around the *SLC30A2* gene, we investigated the expression and regulation of the *SLC30A2*/ZnT2 transporter in control conditions and during specific zinc-feeding regimen in mice (Fig 4). Sizeable expression levels of *Slc30a2* were observed in mouse kidney compared to the intestine and various glandular tissues (Fig 4A). In the mouse kidney, *Slc302*/ZnT2 mRNA expression was the highest in the straight and convoluted parts of the proximal tubule (90–100% normalized expression), compared to much lower expression levels in the collecting duct (16%) and the thick ascending limb (1%) segments (Fig 4B). Immunofluorescence confirmed a strong cytosolic signal for ZnT2 in the cells lining the proximal tubule, positive for the apical receptor megalin (LRP2) (Fig 4C). To investigate the possible regulation of ZnT2 by dietary zinc levels, we applied three types of zinc-feeding regimen during a two-week period (Fig 4D). These dietary manipulations induced significant differences in fecal and urinary zinc excretion, whereas plasma zinc levels remained unaffected. Compared to the control

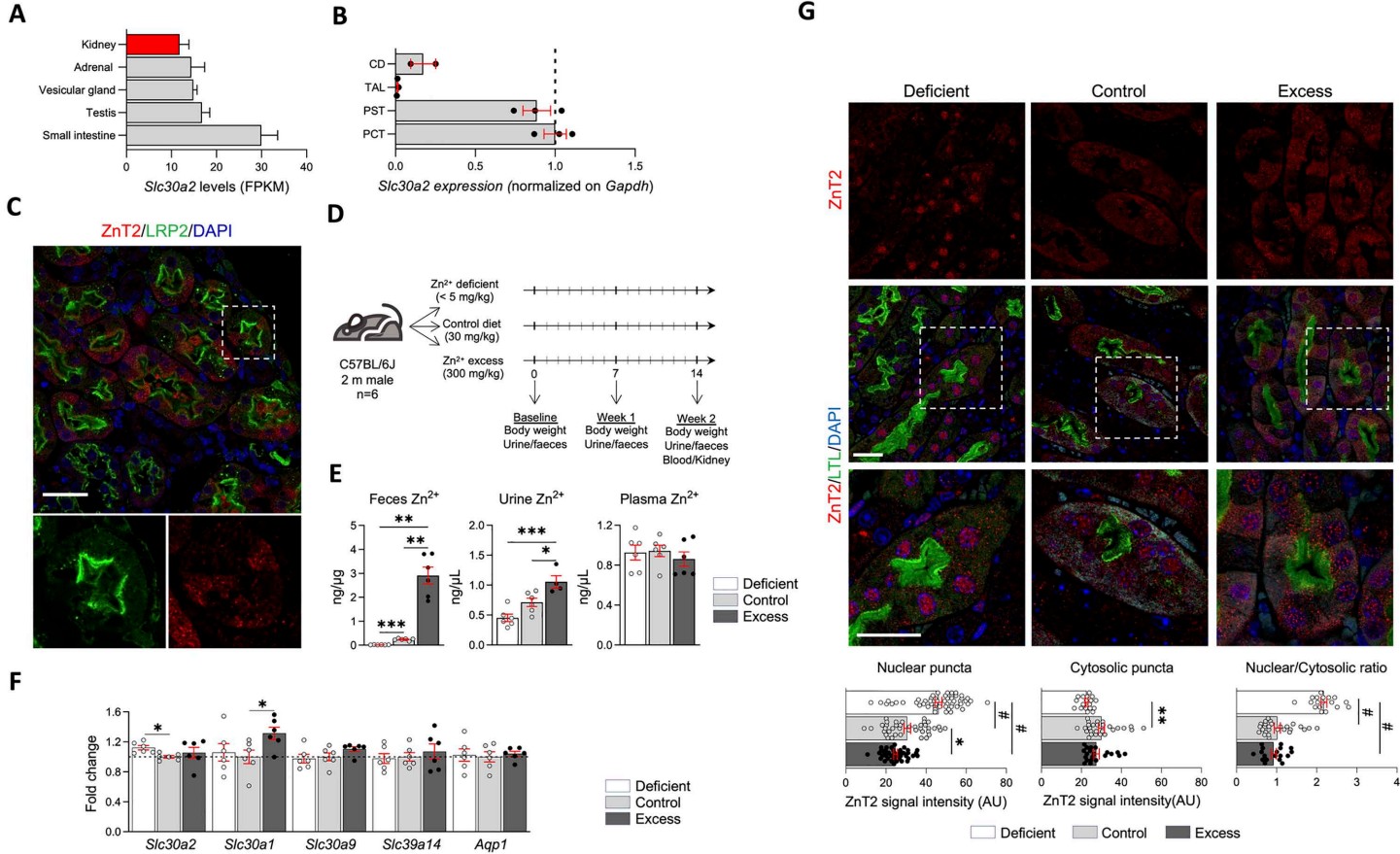

**Fig 4. *Slc30a2* expression in the mouse kidney and dietary zinc modulation. Panel A:** Transcript levels of *Slc30a2* in the five mouse tissues with the highest expression. Data from Li *et al*, *Sci Rep*, 2017. **Panel B:** RT-qPCR analysis of *Slc30a2* levels in isolated tubular segments from mouse kidney. CD, collecting duct; TAL, thick ascending limb; PST, proximal straight tubule; PCT, proximal convoluted tubule. **Panel C:** Co-immunofluorescence analysis of ZnT2 (red) and the proximal tubule apical receptor LRP2 (green). Nuclei were counterstained with DAPI. **Panel D:** Schematics of the dietary intervention protocol. **Panel E:** Measurement of fecal, urinary and plasma Zn2+ levels in the three experimental groups following 2 weeks of specific zinc-feeding regimen (N = 6 animals/group). One-way ANOVA followed by Dunnett's T3 multiple comparison. **Panel F:** RT-qPCR analysis of kidney Zn2+ transporters levels in the three experimental groups, showing that zinc deficiency and excess induced upregulation of *Slc30a2* and *Slc30a1* respectively. In contrast, the expression levels of other proximal tubule Zn2+ transporters and the water channel *Aqp1* were unchanged. **Panel G:** Representative immunofluorescence staining for ZnT2 (red) and lotus tetragonolobus lectin (LTL, green) on mouse kidney sections. Quantification of ZnT2 immuno-fluorescence signal intensity in nuclear puncta, cytosolic puncta, and nuclear/cytosolic ratio is reported on the bottom part of the panel. Bars indicate average ± S.E.M. One-way ANOVA followed by Tukey's Post-hoc analysis: *$P < 0.05$, **$P < 0.01$, ***$P < 0.001$, #$P < 0.0001$.

group (zinc diet: 30 mg/kg), mice exposed to zinc excess (300 mg Zn/kg) displayed highly increased levels of excretion in feces and urine, whereas mice of the zinc deficient group (<5 mg/kg) showed a major decrease in the urine and, to a minor extent, fecal excretion (Fig 4E). A RT-qPCR experiment showed that dietary Zn2+ deficiency resulted in a slight but significant upregulation of the *Slc30a2* mRNA levels, whereas Zn2+ excess did not significantly affect *Slc30a2* expression but was associated with higher levels of *Slc30a1*, another major Zn2+ transporter of the kidney proximal tubule (S4 Dataset). Note that SLC30A1 is localized on the basolateral membrane of kidney tubular cells [33], involved in the protection against Zn2+ toxicity [34] and regulated by dietary zinc levels [35]. These changes contrasted with the overall stability of other Zn2+ transporters including *Slc30a9* and *Slc39a14* and the proximal tubule water channel *Aqp1* (Fig 4F). Using immunofluorescence, we further characterized the subcellular localization of the ZnT2 transporter in the proximal tubule

cells across the three zinc-feeding groups. A significant, dose-dependent shift in the signal for ZnT2 in the nuclear region was observed as a function of zinc exposure (Fig 4G). Relatively to the control diet group, zinc deficient animals showed high ZnT2 levels on the nuclei, co-localizing with DAPI-highlighted nucleic acids, with a significant depletion of the cytosolic signal. In contrast, animals from the zinc excess group did not show any significant shift in ZnT2 localization compared to controls (Fig 4G).

To better elucidate the consequences of the changes associated with dietary $Zn^{2+}$ modulation, we performed RNA-sequencing on mouse kidneys following the two-week feeding regimen. The dietary modulation of $Zn^{2+}$ appeared to have an overall minor impact on the kidney transcriptome, with only 12 differentially expressed genes (DEGs) found in $Zn^{2+}$ deficient kidneys compared to controls (Panel A in Fig P in S1 File), and only three in $Zn^{2+}$ excess animals. Of note, all the upregulated DEGs in $Zn^{2+}$ deficient samples belonged to the interferon signaling pathway, as shown by over-representation analysis (Panel B in Fig P in S1 File). These findings were validated by RT-qPCR (Panel C in Fig P in S1 File), confirming that dietary $Zn^{2+}$ deficiency leads to induction of the interferon signaling cascade, mimicking a transcriptional feature typically observed in bacterial and viral infections [36, 37]. Collectively, our data unveil a complex interplay between dietary zinc levels, kidney transporter dynamics, and immune signaling - suggesting that zinc homeostasis may be a key determinant of kidney function and disease susceptibility.

## Ecological correlation at the SLC30A2 locus

The SNPs identified in the GWAS on urinary zinc levels exhibit large geographical variation in allele frequencies across countries, as do global variations in zinc deficiency prevalence. Since the discovered variants have a rather substantial effect on urinary zinc levels, we hypothesised that regions with higher urinary zinc-increasing allele frequencies may exhibit higher zinc deficiency. Given the available data, this hypothesis could only be tested indirectly using dietary-based zinc deficiency estimates and limited measured serum zinc data, which may not reflect renal zinc handling. To test this hypothesis, we analyzed country-specific allele frequencies and zinc deficiency data. Notable differences in per-country allele frequencies were observed for genetic variants in zinc transporters including rs2272662 (SLC39A4, 33.1%-94.4%), rs807244 (SLC30A2, 1.4%-35.3%), rs3008217 (SLC30A2, 5.3%-43.7%), rs12734494 (SLC30A1, 12.8%-56.1%) as well as for rs2434860 (DPEP1, 36.8%-99.6%) and rs3825417 (RB1/RCBTB2, 14.8%-88.0%; Fig 5A and S5 Dataset). We then correlated the frequency of urinary zinc-increasing alleles with measured and estimated per-country zinc deficiency prevalences, for which data on 13 and 82 countries, respectively, were available. Measured zinc deficiency data came from a review that assessed plasma/serum zinc concentrations from national surveys in low-and middle-income countries [38]. Estimated zinc deficiency prevalences were derived from dietary zinc intake using national food balance sheets, making them more reflective of nutritional factors [1,39,40]. Note that, based on the 13 complete observations, measured and estimated zinc deficiency prevalences were positively correlated (Pearson correlation = 0.46; one-sided p-value = 0.05). Furthermore, we found significant positive correlations with estimated zinc deficiency prevalence (p-value < 0.05) for five of the eleven GWAS SNPs, including variants in *SLC39A4, SLC30A8, SLC30A2, DPEP1*, and *RB1/RCBTB2* (Fig 5A and Table F in S1 File), supporting an overall trend that zinc-increasing alleles correlate with estimated zinc deficiency (one-sided binomial test p-value = 1.12E-04). However, as estimated zinc deficiency prevalence is derived from dietary intake data, these associations may primarily reflect population/ancestry differences in diet, or socio-economic factors rather than direct metabolic responses to the lack of zinc. On the other hand, no significant correlations were observed for measured zinc deficiency prevalence, and any apparent trends (8 positive vs 3 negative). However, this lack of evidence for a correlation should be interpreted cautiously given the very small sample size and the lack of overlap between the genetic basis of urinary and serum zinc measures.

Next, we aggregated the allele frequencies of the eleven SNPs into a per-country genetic risk score (GRS), calculated by multiplying allele frequencies by their effect sizes on urinary zinc levels. Across 82 countries, the GRS was highest

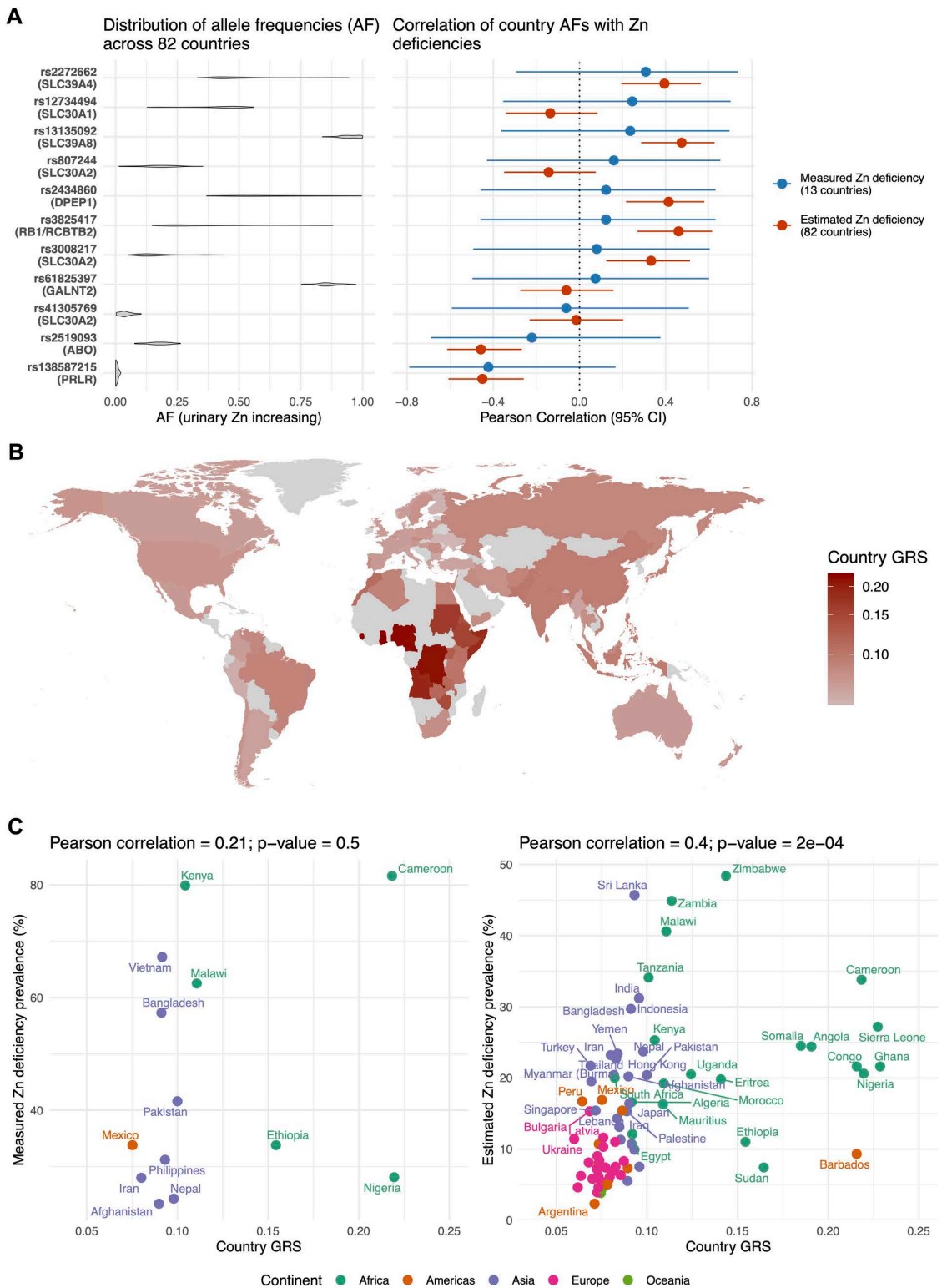

**Fig 5. Ecological correlation analysis between zinc deficiency prevalences and genetic determinants of urinary zinc levels. Panel A:** Distribution of per-country frequencies of urinary zinc increasing alleles and correlations with measured and estimated zinc deficiency prevalences. **Panel B:** Geographical distribution of country genetic risk scores (GRS) defined as the sum of urinary zinc GWAS SNP allele frequencies multiplied by their

effect sizes on urinary zinc levels. **Panel C:** Scatterplot and Pearson correlation between the prevalence of Zn deficiency (measured and estimated) and country GRS. Base map source: the country borders were drawn in R using the public-domain Natural Earth project dataset as distributed with the maps R package ("world" data, https://cran.r-project.org/package=maps).

in sub-Saharan Africa (Fig 5B and S5 Dataset), a region also bearing the highest burden of zinc deficiency. A significant positive correlation was observed between the GRS and estimated zinc deficiency prevalence (Pearson correlation = 0.4, p-value = 2E-04), while a positive but non-significant correlation was found with measured zinc deficiency prevalence (Pearson correlation = 0.21, p-value = 0.5; Fig 5C). Although these results are interesting, their interpretation is very difficult and we provide some hypotheses in the Discussion.

### Effect of SLC30A2 variation on zinc levels in human milk

Previous studies have linked mutations in the *SLC30A2* zinc transporter to insufficient secretion of zinc into human milk [14, 16]. Further, RNA sequencing (RNAseq) data of *SLC30A2* in human mammary epithelium tissue postpartum revealed that *SLC30A2* expression increases post-partum (Fig Q in S1 File). Additionally, single-cell RNA sequencing (scRNAseq) data showed that *SLC30A2* expression is localized to luminal progenitor cells in the breast (Fig R in S1 File). Given these findings, we tested whether the top identified genetic variant was associated with zinc levels in human milk. This analysis was conducted with data from the INSPIRE study, which includes 11 cohorts, six of which are of African ancestry [35, 41, 42] and therefore enriched for the risk allele associated with urinary zinc levels. Zinc levels in human milk and genotype data were available for 387 mothers. Assessing the rs3008428 genotype, which showed the highest genetic correlation with the top *SLC30A2* SNP (rs3008217) among the genotype data in the INSPIRE study, revealed a positive but non-significant association between the risk allele and zinc levels in human milk (beta = 0.035, p-value = 0.50, S6 Dataset).

### Discussion

To our knowledge, this is the first GWAS analysis examining the association with *urinary* zinc excretion. The main genetic signal (p-value = 2.42E-110) localized to a genetic region on chromosome 1 coding for the *SLC30A2*/ZnT2 zinc transporter, which is responsible for $Zn^2+$ ion deposition into secretory granules for subsequent exocytosis [6,43]. Fine-mapping of these loci identified three independent credible sets for the most significant association on chromosome 1, while other regions contained only a single credible set. In total, this resulted in 11 independent signals which we investigated in detail via a colocalization approach, integrating gene expression and protein regulation. While for many loci we found either no or conflicting evidence for colocalization, very robust evidence emerged that the genetic control of urinary zinc levels identified at the 1p36.11 locus is driven via altered expression of *SLC30A2*, a crucial zinc transporter in the kidney.

Notably, we found no similarities in the genetic architecture of zinc homeostasis in blood and urine. Whereas zinc concentration in blood was found to be regulated by genetic factors related to hematopoietic traits [19], our results emphasize the role of zinc transporters (*SLC30A2* in particular) in the kidney in regulating urinary zinc excretion. Our experimental investigations in a murine model of zinc feeding support this observation by showing large increases in urine but not blood zinc levels upon zinc feeding, suggesting that the body maintains stable circulating zinc levels by regulating fecal excretion and adapting its renal handling through urinary elimination, in addition to modulating intestinal absorption. These findings are important considering that there is currently no truly reliable biomarker of zinc status [21,44]. This adaptation is supported by other data showing variability of type and location of zinc transporter depending on zinc status. For example, zinc deficiency promotes accumulation of ZIP4 on the surface membrane of intestinal enterocytes [45]. High extracellular zinc on the other hand triggers lysosomal removal of ZIP4 [46]. Another buffering and regulatory mechanism for zinc are the metallothioneins. They bind zinc intracellularly and serve as a dynamic pool for stabilizing zinc levels [47].

Epidemiological analyses between urinary zinc excretion and cardiometabolic traits replicated several known associations. Zinc deficiency can lead to inflammation and previous studies found low levels of zinc to correlate with inflammatory markers [48,49]. Consistently we observed a correlation between high urinary zinc excretion and CRP levels although no causality could be inferred in the MR analyses. Furthermore, increased urinary zinc excretion was associated with high blood pressure which was previously proposed to be a consequence of hypertension [50–52]. High urinary zinc levels were also correlated with total cholesterol and triglycerides, as well as an increased risk of stroke. However, the MR analysis supported a causal role of urinary zinc excretion only for stroke, not for total cholesterol and triglycerides. Interestingly, the severity of atherosclerosis was previously found to correlate with higher urinary, but not with serum zinc levels [53]. Our study supports the well-known relationship of zinc loss in diabetic patients through a strong epidemiological correlation and a significant reverse MR causal effect. Experimentally-induced hyperglycemia in dogs and rats is known to result in significant hyperzincuria [54–56]. Increased zinc excretion as a consequence of diabetes likely occurs due to hyperglycemia interfering with zinc reabsorption in the renal tubular cells [57,58].

As the detrimental effect of zinc deficiency on health is well-documented, several studies have investigated the impact of supplementation on these conditions. Benefits of zinc supplementation included decreased CRP [59], lipid and glucose levels [60–62] as well as the management of hyperglycemia in diabetes [57]. While the reported links between diabetes, inflammation and zinc were generally concordant, the role of zinc in atherosclerosis and ischemic heart disease has yielded more conflicting results [63].

Since the *SLC30A2* locus has by far the largest impact on urinary zinc levels and the colocalization signal was the clearest for this gene, we focussed on *SLC30A2* and its lead variant (rs3008217). Our ecological correlation analysis showed significant positive correlation between estimated zinc deficiency prevalence and the frequency of the rs3008217-C allele. Its frequency varied considerably across human populations, displaying high allele frequencies in Sub-Saharan Africa (≥40%) [1,30]. Considering that the SLC30A2/ZnT2 transporter is responsible for zinc secretion in human milk, we hypothesized that this correlation may be induced by a possible selection mechanism for offspring survival in regions with elevated prevalence of zinc deficiency, with the rs3008217-C allele leading to an increased zinc secretion in human milk, and thus to greater zinc amounts available for the breastfed child [10]. Alternatively, the frequency of the rs3008217 C may have dropped as a consequence of the out-of-Africa bottleneck. However, the GWAS for human milk zinc concentration in an African sample found a non-significant association of the rs3008217-C allele with zinc levels in human milk, and despite the modest sample size, the effect - even if there is one - is likely to be substantially attenuated compared to that in urine. This hypothesis of natural selection via maternal zinc excretion is supported by Phe-WAS data in European samples, reporting the rs3008217-C (higher urinary zinc excretion) allele being associated with decreased height, trunk, arm, leg, body mass, as well as poorer forced vital capacity (Table G in S1 File).

Interestingly, the ecological correlation seems to hold not only for the SLC30A2 variant, but more globally for the polygenic score for high zinc excretion which might suggest that geographic distribution of genetic determinants driving high urinary zinc may contribute to the observed geographic patterns of zinc deficiency. If the genetic basis of zinc excretion is similar between breast milk and urine, the above-outlined breast milk hypothesis can provide an alternative explanation. Regardless of the underlying causes, populations with inadequate dietary zinc and phytate intake or inadequate intestinal zinc absorption also carry an increased genetic risk for high urinary zinc levels, potentially exacerbating existing nutritional deficiencies. Despite the proposed hypotheses involving natural selection, bottleneck event or adaptation, the observed ecological correlations may well be confounded and simply reflect shared geographic patterns in genetic variation, dietary practices, and socio-economic context.

Further examining publicly available data reporting gene expression in human mammary tissue, we observed increasing *SLC30A2* gene expression in mammary epithelium postpartum (Fig Q in S1 File) [64], whilst single cell RNA expression data further showed presence of *SLC30A2* expression in both mammary gland cells and kidney cells (Fig R in S1 File) [65,66].

We next characterized the influence of established zinc-feeding conditions on the expression and localization of ZnT2 transporter and downstream pathways in mouse models. We observed that ZnT2 is normally expressed in the proximal tubule of the nephron, a segment primarily involved in the regulation of body fluid homeostasis and the processing of filtered nutrients, vitamins and hormones [67]. Exposure of mice to two-week zinc-feeding regimen was reflected by significant, proportional changes in the urinary and fecal zinc excretion levels while plasma zinc levels were comparable across the three groups. Remarkably, ZnT2 expression in the kidney was upregulated in zinc-deficient conditions, with a strong enrichment of the signal in the nuclear region, shifting from the mostly cytosolic distribution observed in the Zinc control and excess groups. This shift was associated with the upregulation of genes involved in interferon signaling – reflecting the established role of zinc on interferon-g signaling and antiviral immunity [68,69]. Under normal conditions, ZnT2 is typically localized to Golgi-derived vesicles near the nucleus [70] and functions to sequester zinc into intracellular vesicles to prevent toxicity [43]. Given zinc's crucial role in DNA and RNA metabolism, it is plausible that zinc deficiency induces a marked relocation of ZnT2-positive vesicles toward the perinuclear membrane [6,71]. In contrast, under zinc-excess conditions, elevated ZnT1 expression may facilitate the excretion of surplus zinc via urine, thereby preventing zinc toxicity [43,72].

This study has several limitations. First, we used data from European cohorts, where the prevalence of dietary zinc deficiency is generally low [1], which precludes determining the exact health-effects resulting from an increased urinary zinc excretion and its associated genetic variants. As a consequence, the ecological analysis is error-prone as it assumes that the effect of the lead variants is similar in European and other populations.

Second, while the colocalization analyses yielded several new and plausible findings, we could not apply the "coloc. susie" (which relaxes the single causal variant assumption) approach for every tested trait due to the lack of appropriate LD reference panel for each of the corresponding summary statistics and studies. Therefore, the colocalization results can be considered as a screening approach. Nevertheless, we report full colocalization results which include probabilities for every tested hypothesis, including PP.H3 (corresponding to the situation when both traits have association in the region but are driven by distinct causal variants), allowing the readers to inspect a particular region of interest in detail.

Third, while we report epidemiological correlations and MR associations between zinc excretion and cardiometabolic traits, these results do not necessarily imply causality and further studies will be needed to corroborate these findings. Although we conducted sensitivity analyses to assess the robustness of the MR results, these analyses rely on strong assumptions that are difficult to verify in practice. On the other hand, null MR effects do not necessarily indicate the absence of a causal relationship, but often may be due to lack of power in the absence of a sufficient number of instruments. Especially, for the forward MR analysis, the statistical power was low due to the limited number of SNPs available to instrument urinary zinc levels.

Fourth, we only tested for linear associations between zinc levels and cardiometabolic traits. While higher zinc excretion in human milk may be beneficial for the offspring, elevated urinary zinc excretion in geographic regions with zinc-deficient diet can be detrimental for health. For other zinc transporters, SLC39A8 and SLC39A4, genetic adaptation has already been demonstrated [73–75]. In a similar fashion, it is likely that an optimal level of zinc excretion exists for SLC30A2, which has likewise been shaped by genetic adaptation, and deviations of urinary excretion in either direction may be disadvantageous to human health. Therefore, future research should examine a possibly non-linear (U-shaped) relationship between urinary zinc concentration and phenotypic outcomes.

Fifth, while the correlation between zinc-increasing allele frequency and zinc deficiency aligns with expectations, the ecological correlation analysis was only borderline conclusive. We believe this is likely due to power limitations: the number of countries with country-specific zinc deficiency estimates is limited, these estimates are likely to be noisy, and the genetic alleles examined are relatively weak predictors of zinc excretion, especially since the polygenic predictor we used is based on European samples only. Furthermore, other potential mechanisms can also give rise to such geographic correlations amongst populations, including different ancestral zinc levels in soils, distinct ancestral diet, divergent socio-economic

conditions or differences in pathogen-induced zinc starvation. More in-depth analysis separating these effects necessitates future research, as was conducted for selenium [76], but it is out of the scope of our current investigation.

Sixth, the lack of clear replication of the *SLC30A2* association with urinary zinc levels in human milk is unlikely to be driven by insufficient power and hence opens the possibility that the genetic regulation of zinc levels in human milk is distinct from that in the urine.

Finally, whilst the experimental approach allowed for characterizing the function of *SLC30A2*/ZnT2 in animal models, we were unable to investigate for the effects of the human-only rs3008217 genetic variant in this particular setting.

To conclude, this study represents the first genome-wide exploration of the genetic regulation of urinary zinc levels, pinpointing key implicated zinc transporters, contrasting genetic pathways governing urinary vs blood concentrations and hinting at possible evolutionary signatures of natural selection to maintain the equilibrium of zinc homeostasis.

## Materials and methods

### Ethics statement

The SKIPOGH study was approved by the ethical committees of the Lausanne University Hospital, the Geneva University Hospitals, and the University Hospital of Bern. All participants provided written informed consent. The CoLaus study was approved by the Ethics Committee of the Canton of Vaud (www.cer-vd.ch; reference PB_201800038, 239/09), and all participants provided written informed consent. The GCKD study was registered in the national registry for clinical studies (DRKS 00003971). The ethical approval from the committees in the involved institutions was obtained, all participants provided written informed consent. The INSPIRE study was approved for all procedures by each participating institution, with overarching approval from the Washington State University Institutional Review Board (protocol #13264). All participants provided written informed consent.

### Study populations

**SKIPOGH.** The SKIPOGH study (Swiss Kidney Project on Genes in Hypertension) is a multicenter family-based population study aiming at exploring genetic and environmental determinants of blood pressure and renal function as well as other health-related cardiometabolic outcomes [29,30]. Study participants were recruited in the city of Lausanne (LS) and the cantons of Geneva (GE) and Bern (BE) between 2009 and 2013 as previously described [30]. Inclusion criteria were: (1) written informed consent, (2) 18 years of age, (3) European ancestry, and (4) at least one first-degree family member willing to participate in the study. Women who reported being pregnant were excluded from the study. At the end of the recruitment period, the study population included 1128 participants of which 1011 have been genotyped. All included participants attended a morning medical visit after an overnight fast, provided blood samples and 24h urine collections, and completed a self-administered questionnaire inquiring about lifestyle habits, socioeconomic circumstances, and medical history. We used 24h urinary zinc excretion levels measured with the inductively coupled plasma mass spectrometry method (ICP-MS) which enables the detection and quantification of multiple trace elements in biological samples [77]. Urinary zinc was measured in SKIPOGH participants with available day *and* night urine collections, and summed to obtain 24h urinary zinc excretion (micrograms/24h).

The SKIPOGH study was approved by the ethical committees of the Lausanne University Hospital, the Geneva University Hospitals, and the University Hospital of Bern.

**CoLaus|PsyCoLaus.** CoLaus|PsyCoLaus is a longitudinal cohort that includes a general population-based sample from the city of Lausanne, Switzerland. The baseline examination started in 2003–2006 with follow-up measures up to 2021. The sample selection method, participants' data collection, samples analysis and cardiovascular events adjudication have been described previously [31]. Briefly, from the city of Lausanne's register 19,830 individuals were selected randomly, representing 35% of the registered population. Inclusion criteria were: (1) written informed consent, (2) aged from 35 to 75 years, and (3) European descent. Measurement of zinc was performed through inductively coupled

plasma mass spectrometry within a nested study of CoLaus|PsyCoLaus, namely ToxiLaus, on 6,447 participants from the baseline survey (2003–2006, Note B in S1 File). ToxiLaus has been thoroughly described by Mouti et al. [78]. The study was approved by the Ethics Committee of the Canton of Vaud (www.cer-vd.ch; reference PB_2018–00038, 239/09), and all participants provided written informed consent.

**GCKD.** The German Chronic Kidney Disease (GCKD) study is an ongoing prospective study that enrolled 5,217 adults with chronic kidney disease between 2010–2012 [32]. The study participants met the following inclusion criteria: age 18–74 years, estimated glomerular filtration rate (eGFR) ranging from 30 to 60 ml/min per 1.73m² or eGFR > 60 ml/min per 1.73 m² with a urine albumin-creatinine ratio (UACR) > 300 mg/g, or a urinary protein to creatinine ratio > 500 mg/g. The participants have been under regular nephrological control. During the initial visit, the biomaterials of participants were collected, frozen, and dispatched to a central biobank, where they were stored at -80°C [79]. Additional details on study structure, procedural guidelines, and characteristics of the recruited cohort have been previously published [80]. The GCKD study was registered in the national registry for clinical studies (DRKS 00003971). The ethical approval from the committees in the involved institutions was obtained, all participants provided written informed consent. For the current project, zinc measurements were quantified through inductively coupled plasma mass spectrometry from spot urine samples.

### Genome-wide association study

Zinc levels were measured by Inductively Coupled Plasma Mass Spectrometry (ICP-MS) as described in Note B in S1 File. In each cohort, a GWAS on log-transformed creatinine-adjusted zinc levels was conducted by adjusting for the following covariates: sex, age, body mass index (BMI), smoking status (non-smokers or past smokers vs current smokers), eGFR, trace element batch and the first five principal components. Genotyping, imputation and data pre-processing prior GWAS analyses are described for each cohort in Note A in S1 File. In the GCKD cohort, urine albumin-creatinine ratio (uACR) and in SKIPOGH, the study center were used as additional covariates. In CoLaus (N = 4,542) and GCKD (N = 4,822), the GWAS was conducted with the SNPTEST software (v2.5.2) in a frequentist, additive model on unrelated individuals. In SKIPOGH (N = 896), batch effects for trace elements were corrected using ComBat [81], which were treated as fixed-effects and familial relations as random-effects by using a kinship matrix generated based on imputed genotypes using PLINK [82]. We then conducted a meta-analysis between zinc GWAS conducted in CoLaus, GCKD, and SKIPOGH using the METAL software [83]. SNPs with a heterogeneity p-value < 1E-08, minor allele frequency (MAF) < 0.01 or with a standard error in the allele frequency difference across the three studies > 0.1 were filtered out.

### Statistical fine-mapping and definition of significant independent hits

We performed statistical fine-mapping of zinc GWAS results to identify and prioritize most-likely causal variants in every significant region. We used the "Sum of Single Effects" method as implemented in the susie R package (v0.12.42) [84]. The LD reference panel was based on the TOPMed-imputed genotyping data from the GCKD cohort, which accounted for approximately half of the individuals included in the meta-analysis. The following parameters were used for susie R fine-mapping: "purity" threshold = 0.1 (any credible set that contained a pair of variables with weaker correlation was filtered out), L = 10 (maximum number of non-zero effects in the susie regression model), maximum number of iterations = 100'000, estimate_residual_variance = F (the residual variance was fixed).

### Omics integration and colocalization analysis

We used coloc R package for genetic colocalization and enumeration approach implemented in "coloc.abf" function [85] which provides posterior probabilities (PP) for 5 hypotheses (H): Both traits have no causal variant in a region (hypothesis H0), only one of two traits has a causal variant (H1 and H2, respectively), both traits have causal variants and they

are distinct (H3), both traits share a common causal variant (H4). We considered positive colocalization results between the traits if PP.H4 was equal or higher than 0.5 (and the sum of PP for other hypotheses was lower than 0.5), and strong evidence for colocalization in case PP.H4 was equal or higher than 0.8. We colocalized zinc associations with various omics and clinical traits from the following studies (each time testing the corresponding region of the respective summary statistics):

- Gene expression (eQTL): Human Kidney Atlas, GTEx version 8, and eQTLGen [86–88];

- Protein expression (pQTL): UK Biobank Pharma Proteomics Project (UKB-PPP) and Icelanders proteomics study [89,90];

- Clinical traits and diseases: UK Biobank and FinnGen phenome-wide association studies, CKDGen Consortium Meta-Analysis [91–93].

In addition to the enumeration approach, we used a combination of fine-mapping and colocalization ("coloc.susie") to relax the single causal variant assumption [94]. This approach was applied specifically to the zinc-associated loci where our fine-mapping approach identified more than one credible set (i.e., there were two or more independent signals).

### Comparison of urinary zinc excretion and plasma zinc concentration

In the SKIPOGH cohort, plasma zinc concentration was available in addition to urinary zinc excretion and measured through ICP-MS (Note B in S1 File) [95]. We then calculated the Pearson correlation between urinary and plasma zinc adjusted for sex, age, eGFR, study center and urine albumin-creatinine ratio.

The genetic correlation between the urinary and blood zinc meta-GWAS [19] was calculated using the GenomicSEM R package (v0.0.5c, LD-score regression model) [96]. Bidirectional MR effects were calculated as outlined in the Method section "Mendelian randomization analysis".

### Observational associations with clinical traits

We calculated observational associations between urinary zinc levels and quantitative and binary (i.e., disease status) clinical phenotypes. All associations were adjusted for sex, age, BMI (unless specified otherwise), smoking status (non-smokers or past smokers vs current smokers) and trace element batch. Prior computing the observational associations, zinc concentrations and quantitative clinical phenotypes were log-transformed and standardized to have a mean of 0 and standard deviation of 1. Binary associations with disease status were derived from logistic regressions adjusted for the same covariates. For stroke, we additionally conducted a time-to-event analysis using a Cox regression model. Individuals were right censored either by death (could coincide with stroke) or follow-up time. Observational association analyses were conducted for all individuals with available zinc and phenotype measures in the CoLaus cohort.

### Mendelian randomization analysis

We conducted bidirectional MR analyses to estimate the causal effect of zinc on clinical phenotypes (forward MR) as well as the causal effect of clinical phenotypes on Zn (reverse MR). We followed a two-sample MR study design by using the zinc meta-GWAS and GWAS of clinical phenotypes (Table B in S1 File). First, genome-wide significant SNPs (p-value < 5E-08) from the exposure GWAS were selected and harmonized with the outcome GWAS. SNPs were then pruned based on LD ($r^2 < 0.01$) using the –clump command in PLINK [82]. To mitigate reverse causality, we applied Steiger filtering by removing SNPs with larger outcome than exposure effects (p-value < 0.05) [97]. MR effect estimations were calculated through the inverse-variance weighted (IVW) method with the TwoSampleMR R package (v0.5.6) [98]. To assess the robustness of the results we also applied the mode- and median-based MR methods, MR Egger and

calculated Cochran's heterogeneity Q-statistics, and further calculated the F-statistics to quantify the association strength of the instrumental variables with the exposure using the Cragg-Donald formula [99]. All genetic effect sizes were standardized and MR causal effects are reported on an SD/SD scale (i.e., how many SD change in the outcome is caused by 1 SD change in the exposure).

Furthermore, we conducted sensitivity analyses where we left out the instrumental variable (IV) associated with *SLC39A8* and *ABO* alone, as well as together, as these IVs were observed to be particularly pleiotropic.

## Ecological correlation analysis

Following the identification of SNPs associated with urinary zinc levels, we further examined the correlations between the allelic frequencies of the identified markers and the prevalence of zinc deficiency across various countries. Allele frequencies for the 11 independent SNPs were extracted from the UK Biobank for countries where at least 50 individuals (after quality control) indicated the country as their country of birth [38,91]. Participants with discrepancies between reported and inferred gender, excess relatives, or who had withdrawn their consent by April 2023 were excluded from the analysis. Using the country-specific allele frequencies, we calculated a country genetic risk score (GRS) for urinary zinc levels. The GRS was defined as the sum of the SNP allele frequency (AF) multiplied by the SNP effect size (beta) on urinary zinc levels: $GRS = \sum AF_i \cdot beta_i$ where allele frequencies and effect sizes were harmonized to the same allele. Measured zinc deficiency prevalences were obtained from Gupta et al. (2020), which reported prevalence rates for low- and middle-income countries based on plasma/serum zinc concentrations [38]. We used zinc deficiency prevalences for adult women (non-pregnant, non-lactating) as this data was more complete than for men (values were reported for 18 countries). Estimated zinc deficiency prevalences were obtained from Wessells et al. (most recent estimates from 2005), which used national food balance sheet data, estimated absorbable zinc content of the national food supply, and demographic data [1,39,40]. We then computed the Pearson correlation between allele frequencies/GRS and zinc deficiency prevalences for countries available in both databases. One-sided Pearson correlation p-values were reported since we expected to see a positive correlation (more excretion, more deficiency).

## Zinc specific diets in mice

The effects of specific zinc diets were investigated in wild-type C57BL/6J male mice, according to established protocols [100,101]. Briefly, groups of six mice (age 8 weeks) were assigned to: (1) a control diet (30 mg Zn/kg), (2) a zinc deficient diet (< 5 mg/kg), or (3) a zinc excess diet (300 mg/kg) for two weeks, monitored by body weight and collection of urine and feces. By the end of the two-week period, mice were sacrificed, and blood and kidneys sampled and prepared for immunohistochemistry (protein staining), qRT-PCR (RNA quantification), Western Blotting (protein detection/quantification), and ICP-MS (zinc quantification). A detailed description of the laboratory protocols is available in Note D in S1 File.

## Human milk analysis

The samples and genotypic data used in these analyses were collected as part of the INSPIRE study that has been described previously [42]. Briefly, this study was a cross-sectional, observational study in which milk (n = 410) and saliva (n = 405) samples were obtained from healthy, breastfeeding women between 2 weeks and 5 month postpartum living in 11 international cohorts. The cohorts consisted of 2 European (Spanish and Swedish), 1 South American (Peruvian), 2 North American, and 6 sub-Saharan African (rural and urban Ethiopian, rural and urban Gambian, Ghanaian, and Kenyan) populations. The study was approved for all procedures from each participating institution, and the overarching approval was obtained from the Washington State University Institutional Review Board (13264). Details of the DNA extraction and genotyping have been described previously [41,42]. Associations between the milk zinc concentrations and SNPs on the Illumina Multi-Ethnic Global-8 v1.0 array (MEGA) were evaluated using the efficient mixed-model association eXpedited

(EMMAX) method [102] as implemented in SNP & Variation Suite (SVS; vs Win64 8.10.0; Golden Helix, Bozeman, MT) with an additive genetic model.

Zinc levels were measured in human milk samples as described in Note B in S1 File.

## Supporting information

**S1 File. Supplementary notes A-D, supplementary figures A-R and supplementary tables A-G.**
(PDF)

**S1 Dataset. Finemapping results of the SLC30A2 locus in individual cohorts.**
(XLSX)

**S2 Dataset. Finemapping results.**
(XLSX)

**S3 Dataset. Colocalization results.**
(XLSX)

**S4 Dataset. List of RT-qPCR primers.**
(XLSX)

**S5 Dataset. Country-level zinc deficiency prevalences and allele frequencies.**
(XLSX)

**S6 Dataset. Individual-level, coded data of the rs3008428 genotype and human milk zinc concentrations from the INSPIRE cohort.**
(CSV)

## Acknowledgments

We are grateful for the willingness of the patients to participate in the GCKD study. The enormous effort of the study personnel at the various regional centers is highly appreciated. We thank the large number of nephrologists who provide routine care for the patients and collaborate with the GCKD study. We thank Drs. Eric Olinger and Tomoaki Takata for their support in generating preliminary data. For the acknowledgements of the INSPIRE Consortium, we refer to the S1 File. The authors would like to express their gratitude to the participants of the CoLaus|PsyCoLaus, SKIPOGH, GCKD and INSPIRE population-based studies and to all the investigators who have contributed to the realization of this project.

## Author contributions

**Conceptualization:** Jean-Pierre Ghobril, Katalin Susztak, Julien Vaucher, Aurélien Thomas, Olivier Devuyst, Anna Köttgen, Murielle Bochud, Zoltán Kutalik.

**Formal analysis:** Marie C. Sadler, Jean-Pierre Ghobril, Oleg Borisov, Guglielmo Schiano, Eunji Ha, Yong Li, Janet E Williams.

**Funding acquisition:** Michelle K McGuire, Courtney L Meehan, Julien Vaucher, Aurélien Thomas, Olivier Devuyst, Anna Köttgen, Murielle Bochud, Zoltán Kutalik.

**Investigation:** Maïwenn Perrais, Guglielmo Schiano, Dusan Petrovic, Belén Ponte, Menno Pruijm, Daniel Ackermann, Idris Guessous, Silvia Stringhini, Georg Ehret, Tanguy Corre, Bruno Vogt, Pierre-Yves Martin, Halit Ongen, Emmanouil Dermitzakis.

**Resources:** Olivier Devuyst.

**Supervision:** Katalin Susztak, Aurélien Thomas, Olivier Devuyst, Anna Köttgen, Murielle Bochud, Zoltán Kutalik.

**Visualization:** Marie C. Sadler, Jean-Pierre Ghobril, Oleg Borisov, Guglielmo Schiano.

**Writing – original draft:** Marie C. Sadler, Jean-Pierre Ghobril, Oleg Borisov, Guglielmo Schiano, Olivier Devuyst, Anna Köttgen, Murielle Bochud, Zoltán Kutalik.

**Writing – review & editing:** Marie C. Sadler, Jean-Pierre Ghobril, Oleg Borisov, Maïwenn Perrais, Guglielmo Schiano, Dusan Petrovic, Eunji Ha, Belén Ponte, Yong Li, Zulema Rodriguez Hernandez, Menno Pruijm, Daniel Ackermann, Idris Guessous, Silvia Stringhini, Georg Ehret, Tanguy Corre, Bruno Vogt, Pierre-Yves Martin, Halit Ongen, Emmanouil Dermitzakis, Janet E Williams, Brenda M Murdoch, Michelle K McGuire, Courtney L Meehan, Sébastien Lenglet, Katalin Susztak, Julien Vaucher, Aurélien Thomas, Olivier Devuyst, Anna Köttgen, Murielle Bochud, Zoltán Kutalik.

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
