## [Decision Letter · Decision Letter 0]

23 Jul 2025

PGENETICS-D-25-00560

Genetic determinants of zinc homeostasis and its role in cardiometabolic diseases

PLOS Genetics

Dear Dr. Sadler,

Thank you for submitting your manuscript to PLOS Genetics. After careful consideration, we feel that it has merit but does not fully meet PLOS Genetics's publication criteria as it currently stands. Therefore, we invite you to submit a revised version of the manuscript that addresses the points raised during the review process.

Please submit your revised manuscript within 30 days Aug 22 2025 11:59PM. If you will need more time than this to complete your revisions, please reply to this message or contact the journal office at plosgenetics@plos.org. Please include the following items when submitting your revised manuscript:

We look forward to receiving your revised manuscript.

Kind regards,

Heather J Cordell

Academic Editor

PLOS Genetics

Santhosh Girirajan

Section Editor

PLOS Genetics

Aimée Dudley

Editor-in-Chief

PLOS Genetics

Anne Goriely

Editor-in-Chief

PLOS Genetics

**Journal Requirements:**

At this stage, the following Authors/Authors require contributions: Marie C. Sadler, Jean-Pierre Ghobril, Oleg Borisov, Maïwenn Perrais, Guglielmo Schiano, Dusan Petrovic, Eunji Ha, Belén Ponte, Yong Li, Zulema Rodriguez Hernandez, Menno Pruijm, Daniel Ackermann, Idris Guessous, Silvia Stringhini, Georg Ehret, Tanguy Corre, Bruno Vogt, Pierre-Yves Martin, Halit Ongen, Emmanouil Dermitzakis, Janet E Williams, Brenda M Murdoch, Michelle K McGuire, Courtney L Meehan, Sébastien Lenglet, Katalin Susztak, Julien Vaucher, Aurélien Thomas, Olivier Devuyst, Anna Köttgen, Murielle Bochud, and Zoltán Kutalik. Please ensure that the full contributions of each author are acknowledged in the "Add/Edit/Remove Authors" section of our submission form.

The list of CRediT author contributions may be found here: https://journals.plos.org/plosgenetics/s/authorship#loc-author-contributions

Potential Copyright Issues:

- Figures 1 and 4. Please confirm whether you drew the images / clip-art within the figure panels by hand. If you did not draw the images, please provide (a) a link to the source of the images or icons and their license / terms of use; or (b) written permission from the copyright holder to publish the images or icons under our CC BY 4.0 license. Alternatively, you may replace the images with open source alternatives. See these open source resources you may use to replace images / clip-art:

- Figure 5. Please (a) provide a direct link to the base layer of the map (i.e., the country or region border shape) and ensure this is also included in the figure legend; and (b) provide a link to the terms of use / license information for the base layer image or shapefile. We cannot publish proprietary or copyrighted maps (e.g. Google Maps, Mapquest) and the terms of use for your map base layer must be compatible with our CC BY 4.0 license.

6) Please ensure that the funders and grant numbers match between the Financial Disclosure field and the Funding Information tab in your submission form. Note that the funders must be provided in the same order in both places as well. State what role the funders took in the study. If the funders had no role in your study, please state: "The funders had no role in study design, data collection and analysis, decision to publish, or preparation of the manuscript.".

**Reviewers' comments:**

Reviewer's Responses to Questions

**Comments to the Authors:**

Reviewer #1: The research detailed in this paper is an impressive and comprehensive investigation into the genetic architecture of urinary zinc levels in individuals of European-ancestry and would be of interest to those in the fields of genetic medicine, public health and population genetics. Sadler et al. highlight the role of of SLC30A2 in the metabolism of zinc and provide valuable insights into how zinc levels may be maintained when dietary levels are deficient. They also attempt to evaluate the genetic risk of individual populations to zinc deficiency given the limited data available.

The manuscript is well-written and presents convincing data but can be improved in terms of 1) greater context of the results and 2) discussion of genetic variation of the SLC30A2 locus at the population level. I have outlined my specific, minor comments below.

1) Zinc is a trace metal micronutrient, meaning that it cannot be synthesised by the diet and instead its levels are completely determined by the diet. It may be beneficial to the reader to understand that deficiencies of zinc (and other micronutrients) may then be caused by either a) malnutrition/poor diet or b) low levels of zinc in soils where dietary foodstuffs are grown. When discussing evolutionary arguments (see comments 4 - 7), this is a key point.

2) At the bottom of page 6: “whilst circulating zinc levels are physiologically important, they are not a reliable indicator of individual zinc status” – please explain why this is

3) They suggest that the body maintains zinc levels by adaptations of the renal handling. The authors may consider comparing this the hierarchy of selenium supply to various organs in cases of selenium deficiency, e.g., referencing work such as Sarangi et al, 2018.

4) There has been a wide body of work exploring genetic adaptation of zinc transporters across global human populations (e.g., Zhang et al., 2015; Roca-Umbert et al. 2022; Engelken et al., 2014) that should be referenced in this manuscript. This literature includes strong evidence of local adaptation in SLC39A8 and SLC39A4 mentioned in the manuscript and should be included.

5) There is some confusion / lack of clarity of the hypotheses described in the “ecological correlation at the SLC30A2 locus” section. The stated hypothesis is that regions with higher frequencies of alleles that increase urinary zinc will be found in regions with higher zinc deficiency. They find only a significant positive correlation between allele frequency and estimated zinc deficiency – this is simply a correlation between allele frequency and dietary habits. For the hypothesis to be true, the authors would have to show a positive correlation between explicitly measured zinc levels (although, I recognise that data is a) limited and b) blood data that the authors show cannot be expected to represent urinary data). It should be clearer throughout this section that the correlation discussed is not reflecting metabolic response to zinc and may simply be reflecting a correlation between ancestry and culture/dietary practices/socio-economic factors.

6) It may also be interesting to calculate the correlation between the measured and estimated zinc deficiency data (with the caveat that zinc levels in the blood may not be informative on zinc deficiency). Deviations between measured zinc levels and estimated zinc levels may indicate differences in metabolism of zinc amongst populations.

7) The authors explore one hypothesis for positive selection on SLC30A2 but fail to mention other, more prevalent hypotheses for genetic adaptation in zinc transporters (see literature mentioned in point 3). They should more thoroughly discuss potential mechanisms for driving allele frequency differences amongst populations (e.g., ancestral zinc levels in soils, ancestral diet, pathogen-starvation ..).

8) The authors should also more explicitly recognise that a) polygenic traits often have more complex architecture in sub-Saharan African populations and b) there is poor portability between GWAS studies conducted on individuals of European-ancestry and individuals of diverse sub-Saharan-African diversity. The GRS calculated is therefore likely a poor approximation of urinary zinc levels in the global areas of highest zinc deficiency.

Reviewer #2: The authors present the first GWAS focused on urinary zinc levels, filling a gap in current zinc homeostasis research. This is a highly promising and well-executed study and the integration of genetics, epidemiology, and experimental models is commendable. I have no major concerns with this manuscript.

Please find some minor comments below

- Consider add F-statistics to Table 4.

- Its not clear why MR egger results are not included.

- Suggest to also add IVW regression lines to supp figure 13-15.

- The authors note strong heterogeneity in effect sizes for rs3008217 GWAS associations across cohorts. It may be worth performing fine mapping prior to meta-analysis for this locus.

**Have all data underlying the figures and results presented in the manuscript been provided?**

Reviewer #1: Yes

Reviewer #2: Yes

PLOS authors have the option to publish the peer review history of their article (what does this mean? ). If published, this will include your full peer review and any attached files.

**Do you want your identity to be public for this peer review?** For information about this choice, including consent withdrawal, please see our Privacy Policy .

Reviewer #1: No

Reviewer #2: No

**Figure resubmission:**
---

## [Decision Letter · Decision Letter 1]

16 Oct 2025

Dear Dr Sadler,

We are pleased to inform you that your manuscript entitled "Genetic determinants of zinc homeostasis and its role in cardiometabolic diseases" has been editorially accepted for publication in PLOS Genetics. Congratulations!

Yours sincerely,

Heather J Cordell

Academic Editor

PLOS Genetics

Santhosh Girirajan

Section Editor

PLOS Genetics

Aimée Dudley

Editor-in-Chief

PLOS Genetics

Anne Goriely

Editor-in-Chief

PLOS Genetics

BlueSky: @plos.bsky.social

Comments from the reviewers (if applicable):

Reviewer's Responses to Questions

**Comments to the Authors:**

Reviewer #1: I am happy with the authors' responses to my comments. A great read!

Reviewer #2: All comments have been addressed

**Have all data underlying the figures and results presented in the manuscript been provided?**

Reviewer #1: Yes

Reviewer #2: Yes

PLOS authors have the option to publish the peer review history of their article (what does this mean? ). If published, this will include your full peer review and any attached files.

**Do you want your identity to be public for this peer review?** For information about this choice, including consent withdrawal, please see our Privacy Policy .

Reviewer #1: No

Reviewer #2: No

**Data Deposition**

http://datadryad.org/submit?journalID=pgenetics&manu=PGENETICS-D-25-00560R1

**Press Queries**

---

## [Editor Report · Acceptance letter]

PGENETICS-D-25-00560R1

Genetic determinants of zinc homeostasis and its role in cardiometabolic diseases

Dear Dr Sadler,

We are pleased to inform you that your manuscript entitled "Genetic determinants of zinc homeostasis and its role in cardiometabolic diseases" has been formally accepted for publication in PLOS Genetics! Your manuscript is now with our production department and you will be notified of the publication date in due course.

With kind regards,

Anita Estes

PLOS Genetics

On behalf of:
